# Non-Invasive Assessment of Left Ventricle Ejection Fraction: Where Do We Stand?

**DOI:** 10.3390/jpm11111153

**Published:** 2021-11-05

**Authors:** Alessandra Scatteia, Angelo Silverio, Roberto Padalino, Francesco De Stefano, Raffaella America, Alberto Maria Cappelletti, Laura Adelaide Dalla Vecchia, Pasquale Guarini, Francesco Donatelli, Francesco Caiazza, Santo Dellegrottaglie

**Affiliations:** 1Division of Cardiology, Ospedale Accreditato Villa dei Fiori, 80011 Acerra, Naples, Italy; a.scatteia@gmail.com (A.S.); angelosilverio1988@gmail.com (A.S.); brauss@libero.it (R.P.); fradestefano@alice.it (F.D.S.); raffaella.america@libero.it (R.A.); guarini@iol.it (P.G.); adrfra@alice.it (F.C.); 2Department of Medicine, Surgery and Dentistry, University of Salerno, 84081 Baronissi (Salerno), Italy; 3Coronary Intensive Care Unit, IRCCS Ospedale San Raffaele, 20132 Milan, Italy; cappelletti.alberto@hsr.it; 4IRCCS Istituti Clinici Scientifici Maugeri, 20138 Milan, Italy; laura.dallavecchia@icsmaugeri.it; 5Chair of Cardiac Surgery, Department of Clinical and Community Sciences, Università degli Studi di Milano, 20122 Milan, Italy; francesco.donatelli@unimi.it; 6Marie-Josee and Henry R, Kravis Center for Cardiovascular Health/ Zena and Michael A, Wiener Cardiovascular Institute, Icahn School of Medicine at Mount Sinai, New York, NY 10029, USA

**Keywords:** cardiac imaging, left ventricular ejection fraction, echocardiography, cardiac magnetic resonance, computed tomography, nuclear cardiology

## Abstract

The left ventricular (LV) ejection fraction (EF) is the preferred parameter applied for the non-invasive evaluation of LV systolic function in clinical practice. It has a well-recognized and extensive role in the clinical management of numerous cardiac conditions. Many imaging modalities are currently available for the non-invasive assessment of LVEF. The aim of this review is to describe their relative advantages and disadvantages, proposing a hierarchical application of the different imaging tests available for LVEF evaluation based on the level of accuracy/reproducibility clinically required.

## 1. Introduction

The left ventricular (LV) ejection fraction (EF) is the preferred measure for the evaluation of LV systolic function. It reflects the amount of blood being pumped out of the LV during each cardiac cycle and is expressed as a percentage, where a value ≥50–55% is considered normal. The accurate measurement of LVEF and volumes is the cornerstone of routine cardiology practice, being crucial for diagnostic definition, prognostic classification, and clinical management. In heart failure (HF), LVEF is used to categorize patients with preserved (HFpEF), mid-range (HFmrEF), or reduced systolic function (HFrEF), as well as to stratify for arrhythmic risk [1]. In valvular heart disease, LVEF, along with LV volumes, helps to define the best timing for surgical intervention [2]. More recently, great attention has been devoted to the evaluation of drug-related cardiotoxicity in patients with onco-hematological diseases, and LVEF is considered the first-line diagnostic parameter [3]. Furthermore, LVEF has been largely proven to hold a fundamental prognostic role. In HFrEF patients, it is considered an independent predictor of outcomes in different models including several clinical characteristics, etiology (ischemic or non-ischemic), age, sex, N-terminal fraction of pro-B-type natriuretic peptide, New York Heart Association class III-IV, and comorbidities. A value of LVEF <35% has been shown to be associated with an increased risk of cardiovascular death and all-cause mortality [4,5].

Although LVEF can be measured during cardiac catheterization by contrast left ventriculography, there are several non-invasive imaging modalities that are routinely used in clinical practice. With every technique, LVEF assessment can be estimated either subjectively by visual evaluation or objectively by quantitative methods. Whenever possible, the preference is to employ quantitative measures to minimize variability and to favor greater precision and accuracy.

Non-invasive imaging modalities for LVEF assessment include:Transthoracic echocardiography (TTE);Cardiovascular Magnetic Resonance (CMR);Cardiac computed tomography (CT); andNuclear cardiology imaging modalities: radionuclide multiple-gated acquisition (MUGA) ventriculography, or ECG-gated single photon emission computed tomography (SPECT).

Of note, when measuring LVEF using different modalities in the same subject, the obtained values may differ due to differences in the methodology applied. Thus, the modalities available for non-invasive LVEF should not be considered interchangeable, and when serial LVEF measures are needed, the same imaging test (ideally interpreted by the same operator) should be consistently used.

The aim of this review is to describe the relative advantages and disadvantages of each non-invasive imaging modality available for LVEF evaluation (Table 1) and to propose a hierarchical application of the different imaging tests available based on the level of accuracy/reproducibility clinically required.

## 2. Imaging Modalities for Non-Invasive Assessment of LVEF

### 2.1. Transthoracic Echocardiography

Because of its wide availability and high safety profile, TTE still represents the main technique used to evaluate LVEF in the clinical arena. LVEF measures can be obtained using various methods, based on the settings of image acquisition (mono-dimensional, two-dimensional (2D) or three-dimensional (3D)) and the geometric assumptions used for calculations (linear measurements, areas or volumes). Currently, the biplane method of disks (modified Simpson’s rule) is the recommended two-dimensional method to assess LVEF [6]. It requires the manual tracing of the endocardial border in the apical four-chamber and two-chamber views, in both end-diastole and end-systole (Figure 1).

These tracings are used by the applied software to divide the LV cavity into a predetermined number of disks (usually 20). This method requires substantial geometric assumption because the entire LV cavity border is not traced. Moreover, the shape of the LV cavity cannot be approximated to any single solid figure, but is considered the sum of a cylinder (at the basal level), a truncated cone (from the level of the mitral valve to the papillary muscles), and another cone (attributed to the cardiac apex) [7]. Although this method is the most widely used for LVEF calculation, it may be highly affected by apical fore-shortening and poor acoustic windows, which can make it hard to clearly distinguish the endocardial border, leading to a wide inter-reader variability (19.7%) [8].

Contrast Echocardiography

An intravenous (i.v.) contrast agent can be administered to improve endocardial border detection during TTE, thus improving the accuracy of LV volumes and EF measurements. Moreover, contrast-enhanced echocardiography was shown to substantially reduce inter- and intra-observer variability (mean percentage of inter-reader variability for LVEF can be reduced from 14.3% to 7.4%) [9]. However, while there is no doubt that the use of contrast enables the acquisition of images of improved quality, there are a few setbacks related to the potential, although rare, life-threatening reactions to the contrast agents (mainly allergic reactions or potential mechanical obstruction of the coronary vessels), and to the availability of the technique, especially due to the inconvenience of an i.v. medication [10].

#### 3D-Transthoracic Echocardiography (3D TTE)

3D TTE is considered the best technique to assess LVEF with ultrasound, as it does not require any geometric assumptions (Figure 2).

Images must be acquired over several heartbeats using specific 3D imaging probes. Unlike 2D, 3D methods are less affected by the shape of LV cavity. When compared to other echocardiographic methods, LVEF evaluation using a 3D modality proved to be more accurate and far less variable [8,11]. However, whereas irregular geometry can be addressed by 3D imaging, limited image quality remains a potential issue with 3D echocardiography and the suboptimal spatial resolution may lead to the incorporation of the trabeculae in the myocardial tracing, hence affecting LV volume measurements [12]. It is also generally characterized by lower temporal resolution than 2D echocardiography. Thus, due to its higher reproducibility and capacity to overcome geometric assumptions and foreshortening, 3D is preferred over 2D echocardiography technique when a precise LVEF evaluation is necessary, such as in the case of eligibility for the implantation of therapeutic devices [13]. When available, 3D TTE is also recommended as the technique of choice for the accurate monitoring of the cardiac effects of chemotherapy [3].

***Pros.*** The negligible harm and lack of ionizing radiation works in favor of TTE, making it an ideal modality for the initial and follow-up evaluation of LV function; additional advantages include its ability to provide real-time heart images together with the technique’s wide availability, portability, and bedside feasibility.

***Cons***. All TTE methods require an acoustic window that allows for the adequate visualization of the blood/endocardial border in all the required LV segments to guarantee accurate tracing and measurements. Obese patients, those with chronic obstructive pulmonary disease, and/or reduced intercostal space will often have poor image quality, which may significantly affect the accuracy of the measurements. Suboptimal image quality is also responsible for limited reproducibility and high inter- and intra-observer variability. Moreover, every echocardiographic technique is strictly dependent on operator experience [14].

### 2.2. Cardiac Magnetic Resonance

In the last few decades, CMR has been established as a robust method for the quantification of LV function and volumes. The high spatial, temporal, and contrast resolution of CMR allows clear identification of the blood–myocardium interface, providing highly accurate measurements of LV function and volumes. Furthermore, CMR has shown superiority over 2D TTE in terms of inter-study reproducibility, required when imaging parameters are to be obtained in serial examinations [15]. In a multicenter comparison study between CMR, 2D and 3D TTE, with and without contrast, Hoffman et al. [8] demonstrated that only contrast 3D TTE can reach CMR reproducibility levels. The volumetric LV assessment in CMR is mainly performed using a stack of 8–12 contiguous short-axis cine images (usually with a slice thickness of 6–10 mm and a slice gap of 0–4 mm) covering the entire LV from the atrioventricular ring to the apex (Figure 3).

The acquisition of cine images with 25–30 phases allows to obtain an adequate temporal resolution to correctly identify the end-systolic and end-diastolic phases. The endocardial border is traced in every slice in both end-diastolic and end-systolic phases, the resulting areas are added together and then multiplied for slice thickness and slice gap providing end-diastolic and end-systolic volumes, stroke volumes, and EF. The tracing of the endocardial contours must exclude the anatomical structures such as the papillary muscles and the trabeculae to obtain the correct calculation of the volumes [16,17]. Many automated or semi-automated softwares are available to speed up the process of volume and EF calculation with CMR. Furthermore, as with 2D TTE, the area-length method can be applied in CMR using two orthogonal long-axis cine images, providing a faster methodology to obtain LV morpho-functional parameters (Figure 4). However, this method is considered less accurate compared to the short-axis one in EF estimation, especially in cardiomyopathies with regional rather than global LV dysfunction [18].

***Pros.*** What contributed to make CMR the standard of care for the assessment of LV volumes and function is the possibility of obtaining a 3D reconstruction of the LV chamber, which frees CMR from the limit of geometric assumptions [17,19], together with the total absence of exposure to ionizing radiation. Moreover, CMR overcomes some of the intrinsic limits of other imaging methods, such as the operator and acoustic-window dependence. In addition, since CMR is a multi-planar method, the acquisition of images is not dependent on the position of the heart within the chest. Finally, no contrast agent is required when CMR is applied solely to calculate LV volumes and EF.

***Cons.*** Drawbacks of CMR are mainly represented by the low availability of dedicated scanners, as well as technicians and physicians with adequate expertise in performing and interpreting cardiac studies. Furthermore, the presence of some medical devices or metallic foreign bodies may contraindicate the execution of a CMR scan in some patients. In a small number of cases, claustrophobia prevents from performing a CMR study. Finally, breathing artifacts and heart rhythm disturbances can prevent the acquisition of good-quality images.

### 2.3. Computed Tomography

CT has recently become more widely used as it allows for accurate, non-invasive evaluation of coronary anatomy and detection of atherosclerotic disease. Additionally, cardiac CT raw data acquired with retrospective gating can be used for the measurement of LV volumes, wall thickness, and the assessment of both global and regional function [20,21]. Full coverage of the cardiac cycle is obtained at the expense of high average levels of radiation exposure compared to prospective gating. However, recent technical developments have made it feasible to acquire cardiac CT studies with relatively lower dose of radiation exposure even when retrospective gating is applied [22]. The use of iodinated contrast is needed to obtain adequate differentiation between the blood cavity and the endocardial borders. By reformatting the acquired volume in the standard cardiac panes, LVEF and volumes can be calculated after appropriate LV segmentation. Many automated or semi-automated software are available to help the reader in this task. Unlike CMR, CT images are obtained during a single breath hold and breathing-related artifacts are uncommon. Nonetheless, image quality may be affected by scanner-related factors (such as temporal and spatial resolution) and patient-related factors (including heart rhythm and cooperation capacity) [23,24,25]. Many studies have compared CT with other non-invasive techniques in the assessment of LV function and volumes. Belge et al. [26] found that CT and CMR provide similar and highly correlated measures of LVEF and volumes, with excellent inter-observer variability for both techniques. Greupner et al. [27] showed that 64-row CT allows accurate and reliable evaluation of global LV function when CMR is used as the reference standard and seems to be superior to invasive ventriculography and echocardiography (even with 3D implementation). Moreover, the ability to detect wall motion abnormalities with CT was proven to be non-inferior to that of invasive ventriculography and echocardiography (Figure 5).

***Pros.*** When compared to CMR, CT showed accuracy in assessing LV volumes and function, with the advantage of combining the assessment of coronary anatomy in one single examination [28].

***Cons.*** Systematic use of cardiac CT for LVEF evaluation may be limited by the need for radiation exposure (higher levels of exposure are caused by retrospective gating used to study LVEF), as well as the need for contrast administration, particularly in the case of poor renal function or contrast allergies. Moreover, a low temporal resolution may lead to under-sampling and therefore an underestimation of LV volumes and function [12].

### 2.4. Nuclear Cardiology


*ECG-Gated Single Photon Emission Computed Tomography (SPECT)*


SPECT represents the procedure most commonly performed for the assessment of myocardial perfusion. It is based on the injection of a radiotracer (typically labeled with technetium-99), which is extracted from the blood by the perfused myocytes and retained for some period of time. Photons are then emitted from the myocardium and captured by a gamma camera, where the information is turned into digital data representing the magnitude of uptake and the location of the emission. The result is the creation of multiple tomograms (or slices) of the heart, with a digital display of the radiotracer distribution throughout the organ. Most commonly, a cardiac SPECT study includes two datasets of image acquisition, including one under resting conditions and another after physical exercise or pharmacologic stress. This test is still widely used for the identification of myocardial perfusion defects, as well as for the study of myocardial viability [29]. Gated SPECT, with the incorporation of ECG gating, allows calculation of LVEF and volumes, simultaneously with the evaluation of myocardial perfusion [29]. All contemporary camera–computer systems incorporate software applications capable of quantitative analysis of LVEF and volumes, based on fully automated (and therefore highly reproducible) detection of the border between the myocardium and the blood (Figure 6) [29]. ECG-gated SPECT showed a very good correlation with CMR in the evaluation of LV end-diastolic volume, but not when end-systolic volume and EF were computed [30].


*Radionuclide Multiple-Gated Acquisition( MUGA) Ventriculography*


MUGA ventriculography has also been used for EF calculation. This technique is based on the labelling of the patient’s red blood cells with technetium-99 and the subsequent evaluation of cell counts in end-diastolic and end-systolic phases, offering the possibility to calculate LVEF. Being characterized by good reproducibility, this technique may be excellent for serial evaluation of EF; however, accurate LVEF measurements using MUGA may present some challenges, including the need for patient collaboration and particular care in performing adequate attenuation correction. Nowadays, it is not routinely performed in many nuclear medicine imaging laboratories [31].

***Pros.*** Advantages of nuclear cardiology in the assessment of LVEF are mainly the high availability of the techniques and the possibility, with gated-SPECT, to obtain data on cardiac function and perfusion in one single exam.

***Cons.*** The main disadvantage is the high dose of radiation exposure, which makes such techniques unsuitable in clinical conditions, requiring repeated measures of LVEF and volumes. Moreover, low temporal resolution may lead to an underestimation of LV volumes [11].

## 3. Assessment of LV Systolic Function beyond LVEF: Myocardial Strain

Although the assessment of LVEF remains one of the main tasks of all cardiac imaging modalities, it is clear that changes in LVEF only occur in the late stages of many cardiac diseases. Moreover, LVEF is a surrogate marker of myocardial contractility and is highly dependent on loading conditions at the moment of evaluation.

Myocardial strain refers to the deformation of a myocardial segment from its initial length to its maximum length, following the different fiber orientations. It is expressed as a percentage. LV longitudinal strain represents the longitudinal shortening from the base to the apex. It is expressed by negative values. Radial strain is the radially directed myocardial deformation towards the center of the LV cavity and represents the LV thickening and thinning motion during the cardiac cycle; radial strain is expressed by positive values. Circumferential strain derives from LV myocardial fibers shortening along the circular perimeter observed on a short-axis view and is consequently represented by negative values [32].

Speckle tracking echocardiography (STE) is considered an accurate and convenient method to assess myocardial strain, as it is easy to perform and largely available [33]. Several CMR techniques have also been developed to measure cardiac muscle deformation. Among them, CMR tissue-tracking has become largely used as it is a post-processing technique that can provide myocardial strain values from traditional cine images already acquired during a standard CMR study for LVEF calculation [34,35] (Figure 7).

The clinical utility of myocardial strain imaging has been increasingly investigated over the past two decades in both ischemic and non-ischemic cardiomyopathies, and in many other cardiac conditions potentially affecting LV function. In every clinical scenario, however, the assessment of longitudinal and circumferential strains appears to be very useful in detecting sub-clinical dysfunction in patients with preserved EF; therefore, these strain calculations should be considered in daily clinical practice [36,37,38,39].

## 4. Multimodality Imaging for LVEF Assessment

Given the availability of many different imaging modalities for the non-invasive assessment of LV function and volumes, each with peculiar strengths and weaknesses, differences in terms of spatial and temporal resolution may lead to variable accuracy levels when using these techniques (Table 2) [40,41,42], while methodological differences (dependent upon the imaging window, need for geometrical assumptions, possibility of 3D acquisitions, etc.) may variably affect inter- and intra-observer reproducibility [14].

High spatial and contrast resolution, excellent intra- and inter-observer reproducibility, and lack of radiation exposure make CMR the current gold standard for the evaluation of LV volumes and function [15,38,43]. On the other hand, CMR is still a relatively expensive technique with limited availability, and it cannot be performed in patients with non-magnetic resonance conditional metal implants. Conversely, TTE is mainly characterized by its high temporal resolution, portability, and wide availability; thus, it remains the first-line examination for the assessment of cardiac function [37,44]. However, the standard Simpson’s method is dependent on geometrical assumptions for the measure of LV volumes and EF, and it is highly affected by the patient’s acoustic window. Contrast echocardiography and 3D TTE have been proven to significantly improve the diagnostic accuracy of the standard 2D TTE, providing values for LV volumes and EF that strongly correlate with those of CMR, but their use is frequently limited mainly because of the need for i.v. access, specific ultrasound probes and expertise [1,9,11].

CT and nuclear cardiology imaging are mainly indicated for EF evaluation in cases where there is a need for the simultaneous assessment of coronary anatomy and myocardial perfusion, respectively. However, both techniques expose patients to a certain dose of radiation and this may preclude their use in cases where repetitive measures of EF are required, or in younger patients.

All the described imaging modalities can provide measurements for LV volumes and EF. However, normal ranges may substantially differ among the different modalities and the same approach should be used when a repeating measurement is required over time [14]. Table 3 shows correlation coefficients between the different imaging modalities in LVEF evaluation, using CMR as the reference standard [45,46].

In addition, what should be interpreted as a “normal EF” may slightly change according to age and sex based on physiological variations, and this should be considered when applying each imaging modality, especially during long-term monitoring [8,46].

It would be ideal, but highly unlikely, that every cardiologist has the same access to all the different imaging modalities and that the cardiac volumes and function of every patient would be assessed in the same way everywhere. Alternatively, based on the premises, a scheme based on the technique’s availability is proposed below. Figure 8 shows a flow chart for hierarchical application of clinically available non-invasive imaging modalities for the comprehensive assessment of LV volumes and systolic function.

In summary, in a routine EF evaluation, standard 2D echocardiography can be enough for a patient’s clinical management. However, in patients in whom the value of EF can drastically change clinical management, i.e., in pre-ICD-CRT implant, cardiotoxicity evaluation, post-heart transplant follow-up, and timing of surgery for valvular heart disease, volumetric- and geometrical-assumption-free techniques should be preferred, with CMR, 3D TTE, or contrast echocardiography chosen based on local availability and expertise. CT and nuclear cardiology should only be considered in conditions where coronary anatomy or myocardial perfusion imaging is also required [47]. Moreover, in addition to LVEF and volumes, the measurement of myocardial strain, easily feasible with both CMR and TTE, should always be considered as it can help in highlighting sub-clinical dysfunction.

## Figures and Tables

**Figure 1 jpm-11-01153-f001:**
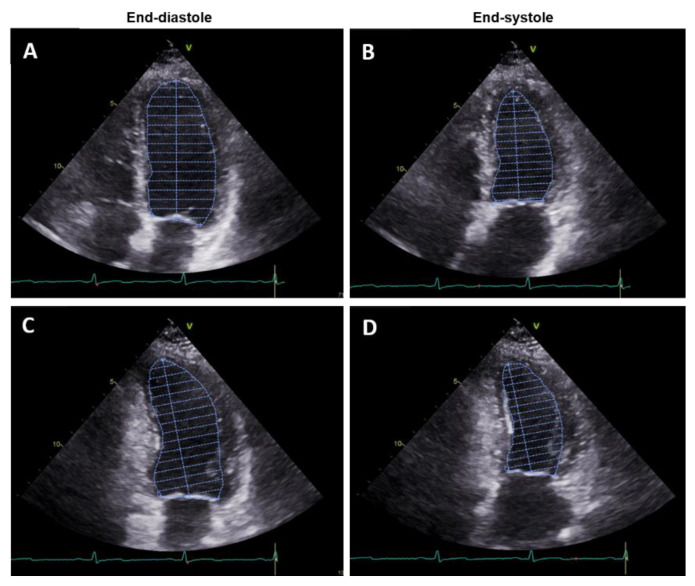
Apical 4-chamber (**A**,**B**) and 2-chamber (**C**,**D**) transthoracic echocardiography in end-diastole (**A**,**C**) and end-systole (**B**,**D**) showing the biplane method of disks for the assessment of left ventricular ejection fraction.

**Figure 2 jpm-11-01153-f002:**
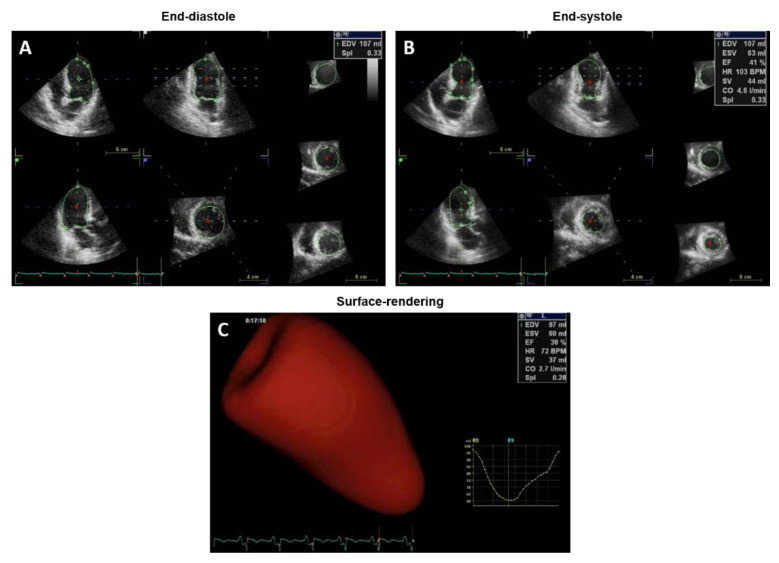
EF calculation by 3D transthoracic echocardiography, endocardial borders are traced in long and short-axis images and a 3D model is reconstructed.

**Figure 3 jpm-11-01153-f003:**
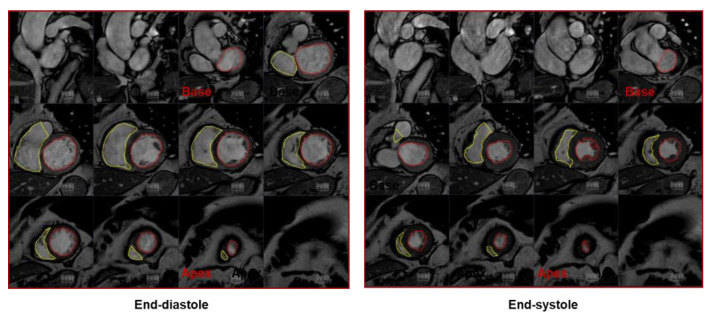
Cardiac magnetic resonance calculation of the left ventricular (LV) ejection fraction (EF): a stack of short-axis cine images covering the entire left ventricle from the atrioventricular ring to the apex. LV (red) and right ventricular (yellow) endocardial contours are drawn in end-diastole and end-systole to measure LV volume and LVEF.

**Figure 4 jpm-11-01153-f004:**
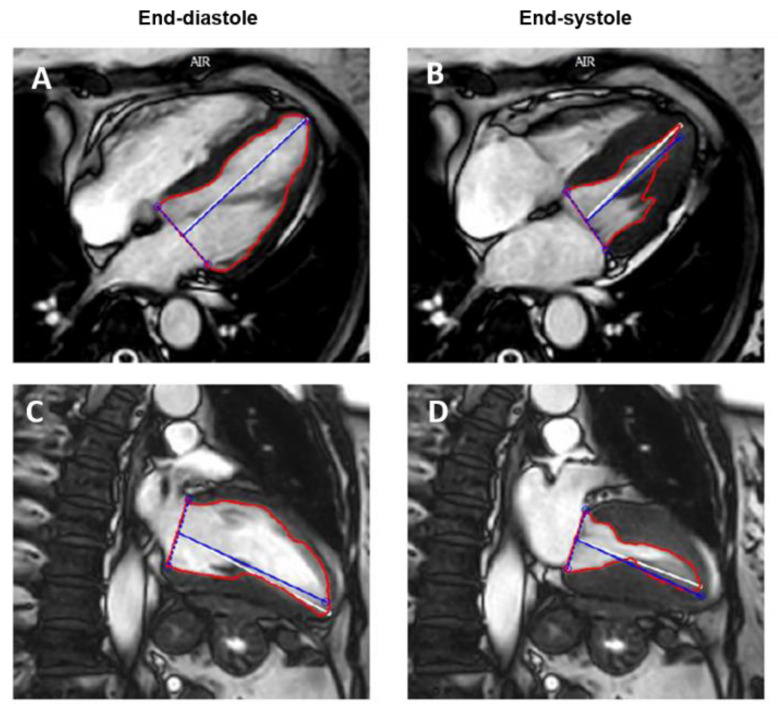
Cardiac magnetic resonance calculation of left ventricular ejection fraction using cine 4-chamber long axis (**A**,**B**) and 2-chamber long-axis (**C**,**D**) images.

**Figure 5 jpm-11-01153-f005:**
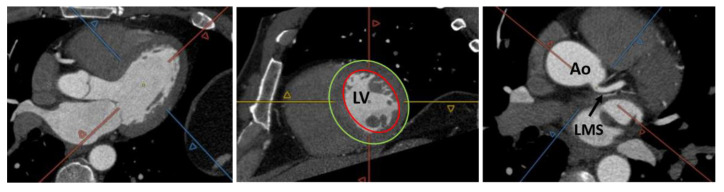
Computed tomography images showing the possibility of combining ejection fraction and coronary artery evaluation. LV = left ventricle, LMS = left main stem, Ao = aorta.

**Figure 6 jpm-11-01153-f006:**
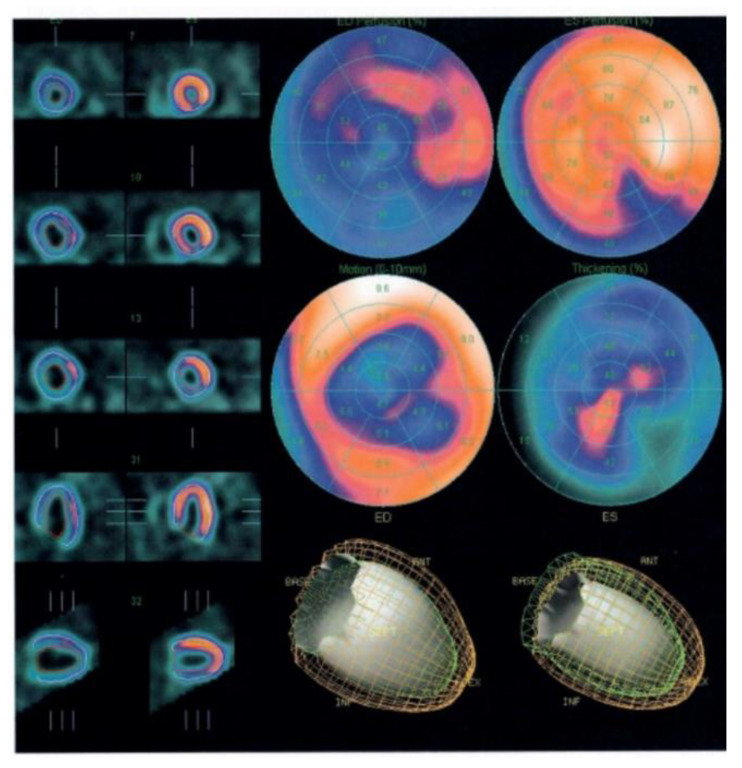
Left ventricular ejection fraction calculated by ECG-gated single photon emission computed tomography. ED = end-diastole; ES = end-systole.

**Figure 7 jpm-11-01153-f007:**
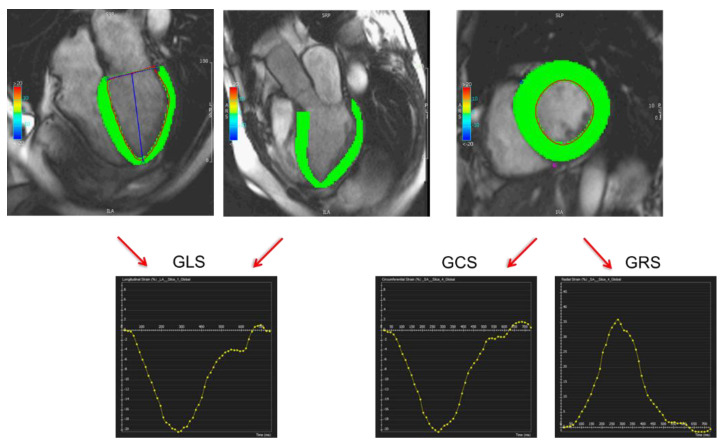
Myocardial strain calculation from cardiac magnetic resonance cine images using feature tracking analysis (post-processing with CVi42, Circle Cardiovascular Imaging Inc., Calgary, AB, Canada).

**Figure 8 jpm-11-01153-f008:**
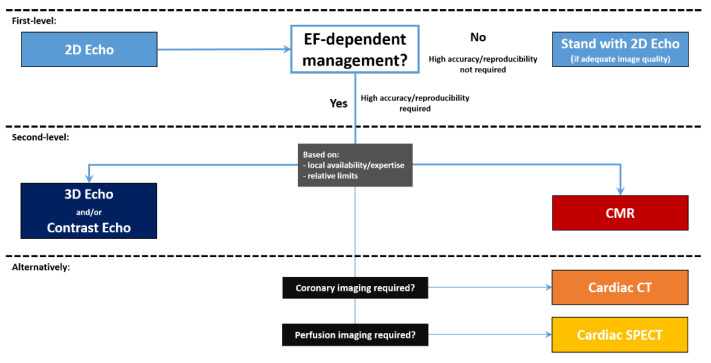
EF evaluation: proposed flow-chart for hierarchical application of clinically available non-invasive imaging modalities. CMR = cardiovascular magnetic resonance, TTE = trans-thoracic echocardiography, SPECT = single photon emission computed tomography, CT = computed tomography.

**Table 1 jpm-11-01153-t001:** Relative advantages and disadvantages of clinically available imaging modalities for LVEF evaluation.

Imaging Modality	Advantages	Disadvantages
**Trans-Thoracic Echocardiography (TTE)**	Wide availability High cost effectiveness Real-time images No ionizing radiation Minimal tissue effects	Depends on acoustic window and operator experience Depends on geometric assumptions
**Contrast-TTE**	Improved accuracy Reduced inter- and intra-observer variability	Low reproducibility Underestimation of volumes in apical fore-shortening
**3D-TTE**	No geometric assumption Improved accuracy	Suboptimal spatial resolution
**Cardiac Magnetic Resonance (CMR)**	No geometric assumption No ionizing radiation High reproducibility High spatial and temporal resolution No contrast administration	High cost Low availability and expertise Non-MR conditional devices
**Computed Tomography (CT)**	High spatial resolution Simultaneous coronary imaging	Ionizing radiation exposure Low temporal resolution Contrast administration
**Nuclear Cardiology Imaging**	Wide availability Simultaneous perfusion imaging	High dose of radiation exposure Low temporal resolution

**Table 2 jpm-11-01153-t002:** Values of spatial and temporal resolution for different cardiac imaging techniques. TTE = transthoracic echocardiography, CMR = cardiac magnetic resonance, CT = computed tomography, SPECT = single photon emission computed tomography [27,30,40].

Imaging Technique	In-Plane Spatial Resolution (mm)	Temporal Resolution (ms)
TTE	0.5–2	15–30
CMR	1–2	20–50
CT	<1	60–165
SPECT	4–15	15–45

**Table 3 jpm-11-01153-t003:** Correlation coefficients between different imaging modalities for the evaluation of left ventricular ejection fraction, using CMR as the reference standard. TTE = transthoracic echocardiography, CMR = cardiac magnetic resonance, CT = computed tomography, SPECT = single photon emission computed tomography.

Imaging Technique	Correlation Coefficient (r^2^) versus CMR
TTE	0.67
Contrast TTE	0.75
3D TTE	0.86
CT	0.52
SPECT	0.67

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
