# Peer review of "Non-Invasive Assessment of Left Ventricle Ejection Fraction: Where Do We Stand?"

_jpm, 2021, doi:10.3390/jpm11111153_

Round 1

Reviewer 1 Report

The authors Scatteia et al. present a comprehensive, concise overview about the currently available techniques for the assessment of left ventricular function. The manuscript is well organized, giving a short overview of the topic and then of each method. Apart from a short description of each method, the autors close each paragraph with a short "pros and cons"-paragraph. In the end, they present a flow chart how to choose the appropriate method based on clinical problems. There are a few minor comments to the authors:

Line 41: Measurement of LVEF during cardiac catherization is an additional method which should be described, not a method "still to be used".

Line 50: The abbreviation MUGA should be placed in parenthesis, not its term.

Please state whether the images are from your own clinic or based on previous work.

Line 70: "respectively" can be omitted.

Line 86: Please add an approximate of the inter-observer variability.

Lines 94/95: Please explaine the conecnrs about the safety of the contrast agents and the terminus "hassle factor".

"CMR" - the authors should consider adding "CMR imaging" or "tomography", corresponding to "CT".

Lines 170: Further "cons" are breathing artefacts and heart rhythm disturbances.

Lines 246: Is it indeed ture that nuclear medicine departments are highly available?

Table 2: Please cite a reference for these values.

Author Response

The authors Scatteia et al. present a comprehensive, concise overview about the currently available techniques for the assessment of left ventricular function. The manuscript is well organized, giving a short overview of the topic and then of each method. Apart from a short description of each method, the autors close each paragraph with a short "pros and cons"-paragraph. In the end, they present a flow chart how to choose the appropriate method based on clinical problems. There are a few minor comments to the authors:

1. Line 41: Measurement of LVEF during cardiac catheterization is an additional method which should be described, not a method "still to be used".

We thank the reviewer for this important clarification; as the review is focused on the non-invasive assessment of LV EF, we decided to not describe the measurement of LVEF during cardiac catheterization in details; however, we removed the “still” from the text.  

2. Line 50: The abbreviation MUGA should be placed in parenthesis, not its term.

We thank the reviewer and the text has been modified accordingly

3. Please state whether the images are from your own clinic or based on previous work.

All the images used have been generated from our own clinic.   

4. Line 70: "respectively" can be omitted.

We thank the reviewer and the text has been modified accordingly

5. Line 86: Please add an approximate of the inter-observer variability

We thank the reviewer for this comment, an approximate value of interreader variability, together with appropriate reference has been added.  

6. Lines 94/95: Please explaine the concerns about the safety of the contrast agents and the terminus "hassle factor". DONE

We thank the reviewer for this comment, the text has been modified accordingly, and the term “ hassle factor” has been changed with a clearer term.

7. "CMR" - the authors should consider adding "CMR imaging" or "tomography", corresponding to "CT".

We thank the reviewer for this comment, however we think that the abbreviations make the text clearer and easier to follow.

8. Lines 170: Further "cons" are breathing artefacts and heart rhythm disturbances.

We thank the reviewer and the text has been modified accordingly

9. Lines 246: Is it indeed true that nuclear medicine departments are highly available?

In our daily clinical practice, it is very easy to get a nuclear imaging test, as there are many nuclear medicine departments in University hospital as well as in many private clinical centers.

10. Table 2: Please cite a reference for these values.

We thank the reviewer and the text has been modified accordingly

Reviewer 2 Report

Scatteia et al. present a review aiming at describing relative advantages and disadvantages of non-invasive measurements methods of LVEF, proposing a hierarchical application of the different imaging tests available based on the level of accuracy/reproducibility clinically required.

The authors should be congratulated for this paper that is relevant, well written and easy to follow.

It displays interesting informations for the reader.

Author Response

We thank the reviewer for this comment and hope the paper will be suitable for publication.